# Effectiveness and cost-effectiveness of basic versus biofeedback-mediated intensive pelvic floor muscle training for female stress or mixed urinary incontinence: protocol for the OPAL randomised trial

Suzanne Hagen,[1] Doreen McClurg,[1] Carol Bugge,[2] Jean Hay-Smith,[3] Sarah Gerard Dean,[4] Andrew Elders,[1] Cathryn Glazener,[5] Mohamed Abdel-fattah,[6] Wael Ibrahim Agur,[7] Jo Booth,[8] Karen Guerrero,[9] John Norrie,[10] Mary Kilonzo,[11] Gladys McPherson,[12] Alison McDonald,[12] Susan Stratton,[1] Nicole Sergenson,[1] Aileen Grant,[13] Lyndsay Wilson[14]

► http://dx.doi.org/10.1136/bmjopen-2018-024152

For numbered affiliations see end of article.

**Correspondence to**
Professor Suzanne Hagen;
s.hagen@gcu.ac.uk

## ABSTRACT

**Introduction** Accidental urine leakage is a distressing problem that affects around one in three women. The main types of urinary incontinence (UI) are stress, urgency and mixed, with stress being most common. Current UK guidelines recommend that women with UI are offered at least 3 months of pelvic floor muscle training (PFMT). There is evidence that PFMT is effective in treating UI, however it is not clear how intensively women have to exercise to give the maximum sustained improvement in symptoms, and how we enable women to achieve this. Biofeedback is an adjunct to PFMT that may help women exercise more intensively for longer, and thus may improve continence outcomes when compared with PFMT alone. A Cochrane review was inconclusive about the benefit of biofeedback, indicating the need for further evidence.

**Methods and analysis** This multicentre randomised controlled trial will compare the effectiveness and cost-effectiveness of PFMT versus biofeedback-mediated PFMT for women with stress UI or mixed UI. The primary outcome is UI severity at 24 months after randomisation. The primary economic outcome measure is incremental cost per quality-adjusted life-year at 24 months. Six hundred women from UK community, outpatient and primary care settings will be randomised and followed up via questionnaires, diaries and pelvic floor assessment. All participants are offered six PFMT appointments over 16 weeks. The use of clinic and home biofeedback is added to PFMT for participants in the biofeedback group. Group allocation could not be masked from participants and healthcare staff. An intention-to-treat analysis of the primary outcome will estimate the mean difference between the trial groups at 24 months using a general linear mixed model adjusting for minimisation covariates and other important prognostic covariates, including the baseline score.

**Ethics and dissemination** Approval granted by the West of Scotland Research Ethics Committee 4 (16/

### Strengths and limitations of this study

► The intervention being evaluated is highly relevant to UK National Health Service practice.
► The primary outcome measure is woman-centred, focusing on the severity of their urine leakage symptoms and the impact on their quality of life.
► An extensive longitudinal qualitative case study and process evaluation will provide rich data about women's experiences of the intervention.
► All trial participants receive a high standard of pelvic floor muscle training, therefore the effect of adding biofeedback will need to be large to show between-group differences.
► Women randomised to the pelvic floor muscle training group could potentially access biofeedback, as devices are available to buy.

LO/0990). Written informed consent will be obtained from participants by the local research team. Serious adverse events will be reported to the data monitoring and ethics committee, the ethics committee and trial centres as required. A Standard Protocol Items: Recommendations for Interventional Trials checklist and figure are available for this protocol. The results will be published in international journals and included in the relevant Cochrane review.

**Trial registration number** ISRCTN 57746448; Pre-results.

## INTRODUCTION

Urinary incontinence (UI), defined as any involuntary loss of urine,[1] is a common condition in women. The prevalence of UI depends on the definition: using a broad definition, a range of between 5% and 69%

is reported, with most studies in the range 25% to 45%.[1] The main types of UI are stress, urgency and mixed, with stress being most prevalent (accounts for around half of all UI), followed by mixed UI (stress and urgency combined) (most studies report 7.5%–25%) and fewer having urgency UI alone (most studies report 1%–7%). The cost to the UK National Health Service (NHS) annually of treating clinically significant UI in women was estimated in the Leicestershire MRC Incontinence Study[2] as £233million, not including the personal costs borne by the women which were estimated to be £178million in the same study.[3] UI significantly impacts on daily living for the majority of women, with an associated increased prevalence of major depression.[4] Among older women, social isolation, psychological distress[5] and increased risk of admission to long-term care institutions[6] feature highly. Thus UI is prevalent and costly to women, the NHS and society both financially and in terms of physical and mental well-being.

Currently, supervised pelvic floor muscle training (PFMT) of at least 3months' duration is the first-line treatment for stress and mixed UI.[7] PFMT involves the regular practice of repeated voluntary pelvic floor muscle contractions, with sufficient exercise progression, in order to produce a training effect on the muscles. The aim of a PFMT programme is to increase the strength of the muscles, to build up muscle volume and thus improve structural support; to increase contraction endurance; to improve muscle resting tone; to improve muscle recruitment through improved nerve function and properties of muscle fibres and improve cognitive awareness of body posture and a relaxed versus an unrelaxed state of the pelvic floor. In addition, a PFMT programme involves instruction on contracting and relaxing the pelvic floor muscles to improve coordination and functional use (termed the 'Knack') to occlude the urethra during physical activities that increase abdominal pressure and precipitate UI. Pelvic floor muscle contraction can also be used to inhibit a bladder contraction at the time of urgency as a method of urge suppression.

To produce improvements in muscle strength and endurance, the basic physiological principles must be adhered to[8]: overload (muscles need to perform more work than usual to the point of fatigue), specificity (muscles must be trained with exercise or physical activity that replicates as closely as possible the functional movement required), maintenance and reversibility (benefits of the exercise will reverse if they are not undertaken on a regular basis). There is evidence to suggest that for effective resistance training in skeletal muscles in adults, two to four sets of 8–12 slow to moderate velocity, moderate to maximal intensity contractions be performed per day, 2–3 days per week with progression over 16 weeks.[9 10] Biofeedback is commonly used as an adjunct to PFMT. Biofeedback is the technique by which information about a normally unconscious physiological process is presented to the patient and/or the therapist as a visual, auditory or tactile signal.[11] Electromyography (EMG) is the study of

minute electrical potentials produced by depolarisation of muscle membrane.[12] In EMG biofeedback, electrical activity arising from muscle activity (during exercise and voluntary effort) is recorded in microvolts and displayed as a visual or auditory signal for both patient and therapist to view. Within a PFMT programme, an internal vaginal probe is used to record electrical information from pelvic floor muscles through surface recordings. The probe is connected by cables to a biofeedback unit. Handheld units with a small visual display screen are available for home use. The display provides a visual representation of the muscles contracting and relaxing, allowing monitoring of strength, endurance and repetitions. The effectiveness of using biofeedback to augment PFMT for UI in women has been the subject of a Cochrane review, the conclusions of which were that women who had PFMT including biofeedback were more likely to report cure or improvement of UI than those who had PFMT without biofeedback, but this finding was potentially confounded by more health professional contact and supervised treatment time in the biofeedback groups.[13] Biofeedback in general is considered an evidence-based behaviour change technique (BCT) according to the BCT taxonomy.[14]

The effect of PFMT relies first on sufficient exercise being undertaken for a long enough period to strengthen and hypertrophy the muscles, and second the continuation of sufficient exercise so that strength is maintained.[15] Thus to intensify PFMT, adherence must be maximised both initially (ie, uptake or adoption of the exercise programme) and then continued over the longer-term (maintenance). Based on current evidence, it is not clear if the apparent benefit of biofeedback can be attributed to the biofeedback or to some other variable such as more health professional contact in those women receiving biofeedback.[13 16] A common problem in previous trials was the failure to clearly state the purpose of biofeedback, or to describe the intervention protocol.[13] Thus, it was not clear if the way the biofeedback was used could theoretically or in practice qualitatively change the intensity or effectiveness of the PFMT. In our trial, the purpose of biofeedback is twofold: to enhance uptake initially as women will know they are performing the correct technique, and to support ongoing exercise training adherence over the longer-term.

The aim of this trial is to determine the effectiveness and cost-effectiveness of PFMT compared with biofeedback-mediated PFMT for the treatment of stress or mixed UI. Both trial groups will have the same PFMT programme and same amount of health professional contact to establish if the addition of biofeedback improves incontinence outcomes.

## METHODS AND ANALYSIS

This trial design is a parallel group multicentre randomised controlled trial, comparing effectiveness and cost-effectiveness of PFMT versus biofeedback-mediated PFMT for women with stress or mixed UI. It is set in UK

# THE OPAL TRIAL

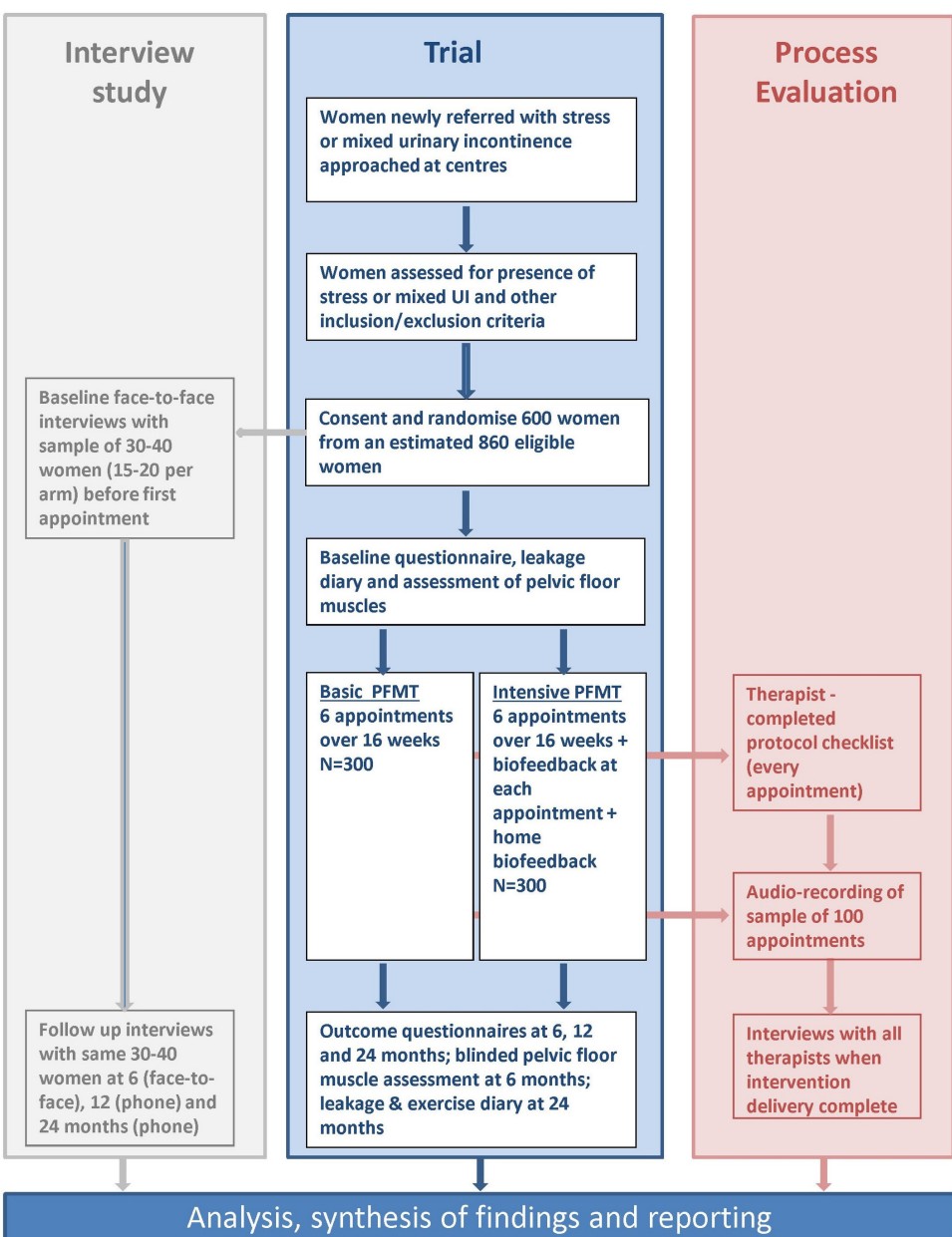

**Figure 1** OPAL study overview. PFMT, pelvic floor muscle training; UI, urinary incontinence.

community, outpatient and primary care settings, where continence care is usually provided. The trial includes a nested longitudinal qualitative case study and process evaluation (figure 1). The trial is assessing superiority of PFMT plus biofeedback over PFMT alone. The start date is 1 September 2013 and finish date 30 November 2018 (figure 2).

## Trial groups

Participants in this trial will be randomly assigned to either PFMT or biofeedback-mediated PFMT. Allocation will be generated by a web-based programme developed and implemented by The Centre for Healthcare Randomised Trials (CHaRT), situated remotely from the trial office and recruiting centres.

Prior to randomisation, as part of the screening process, a visual and digital assessment of the woman's vagina and pelvic floor muscles will be carried out to provide accurate knowledge of the condition of the perineum and vagina (including areas of pain or increased tone), and the woman's capacity for contracting and relaxing her pelvic floor muscles.

## Control group (PFMT)

After randomisation, women will be taught by a trained therapist (nurse or physiotherapist) the correct exercise technique and this will be confirmed on digital palpation with the woman in the supine position. Women will be encouraged to become aware of the sensation or feeling

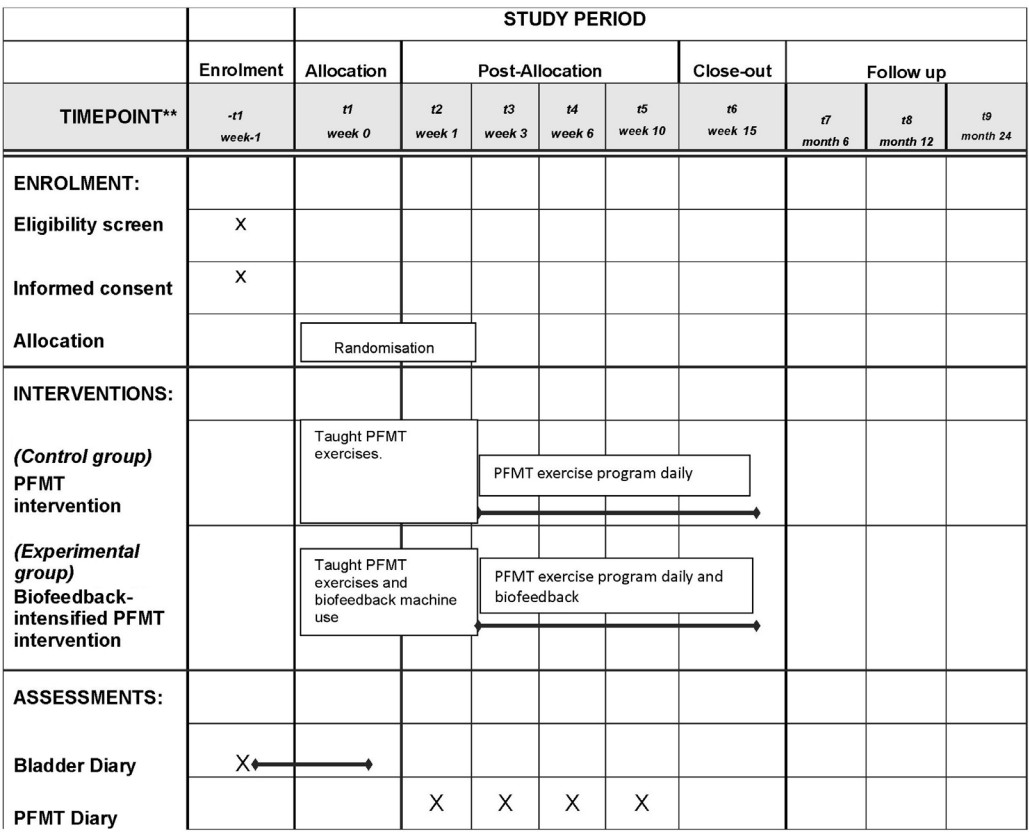

**Figure 2** OPAL SPIRIT diagram. PFMT, pelvic floor muscle training; SPIRIT, Standard Protocol Items: Recommendations for Interventional Trials.

of contracting and holding the appropriate muscles, and also of relaxing them.

Women in the PFMT group will be offered six appointments with the trained therapist over a 16-week period at weeks 0, 1, 3, 6, 10 and 15. During these appointments, the PFMT programme will be delivered and home exercise prescribed.

The initial PFMT programme will be identified and agreed between the individual woman and her therapist over the first and second appointment, according to the woman's ability. The exercise programme will be practised during the appointments to allow the therapist to assess progress and adjust the programme as necessary. Home exercise will be prescribed after the first appointment, or as soon as the therapist confirms a correct technique has been achieved.

There are two main components to the PFMT protocol: resistance training and counterbracing (also called 'The Knack'.[17] Initial exercise prescription in terms of number and type of contractions, length of hold, number of sets, position, will be informed by the PERFECT (Power, Endurance, Repetitions, Fast, Every Contraction Timed) assessment.[18] The starting dose comprises the number (and duration of hold) of near maximal contractions a woman is able to complete before the pelvic floor muscles reach fatigue. The programme is progressed (eg, increasing the number of repetitions by one, or duration of hold by one second, each week) according to

the woman's ability to reach her individualised goal. For example, a goal might be to exercise 3 days per week. On these days the women could aim for three exercise sets, with 3 min between sets. Each set could comprise 10 repetitions held for 10 s each, with 10 s rest between, followed by 10 fast contractions. Counterbracing is taught with a cough—a precontraction of the pelvic floor muscles to be held until the abdominal/diaphragmatic contraction ceases—and over time progressed to include other functional activities as clinically indicated. BCTs are built in to the PFMT protocol. The main categories of BCTs included are: beliefs, emotions and information; teaching and confirming PFM contraction, practising skills; goal setting; action planning; problem solving. A set of 30 core BCTs are to be used if possible at each appointment (eg, agree and record PFMT action plan in new exercise diary), and a further set of 17 optional BCTs are to be used at the therapist's discretion (eg, discuss pros and cons of doing PFMT). The use of the BCTs is recorded by therapists in a checklist.

No biofeedback equipment is used in the control group, although verbal feedback based on digital vaginal palpation is permitted.

### Experimental group (PFMT plus EMG biofeedback)
The use of EMG biofeedback is included in the experimental group in addition to all that is described above for the control group (both PFMT and BCT protocols).

Hand-held EMG biofeedback devices, which measure and monitor pelvic floor muscle contractions, are used to deliver biofeedback. The same type of device is used throughout the trial, both in clinic and at home.

Participants randomised to receive PFMT plus biofeedback will have a biofeedback protocol incorporated into their PFMT appointments, and will be given a biofeedback unit to use during home exercise sessions between appointments. The content of the biofeedback protocol is underpinned by the information-motivation-behavioural skills model of behaviour change,[19] incorporating social cognitive theory,[20] an evidence-based theory relating to self-efficacy and perseverance. It is hypothesised that self-efficacy for PFMT is enhanced because biofeedback offers information to women about a normally 'hidden' muscle activity. Biofeedback also supports motivation through providing audio and visual feedback (graphs on screen which can be printed off) on the 'effort/performance' of the muscle contraction, which facilitates tracking changes in muscle strength and performance. In turn this enhances behavioural skills through improving the performance of a muscle contraction, and the timing of a contraction, to reduce leakage with increases in intra-abdominal pressure (eg, during cough, sneeze, lift). In this way the biofeedback 'intensive' trial group incorporates BCTs additional to those in the control group, that aim to enhance self-efficacy.

### Training in intervention delivery

All therapists delivering the trial interventions are clinical specialists or advanced practitioners already working in the area of continence and will receive training from the trial team on using the biofeedback units and associated software, implementing the PFMT, BCT and biofeedback protocols and completing the trial paperwork to ensure standardisation across centres. An intervention training manual will be provided to each centre covering details of the intervention development and delivery for both the control and experimental groups. Practical sessions using the biofeedback units will be incorporated into the training and there will be a forum for discussion with opportunities for questions. All centres will use the same model of biofeedback unit for clinic and home use.

### Data collection

Data will be collected via participant-completed questionnaires at baseline, 6, 12 and 24 months. In addition, at baseline women will record their leakage episodes in a 3-day bladder diary. During the 16-week intervention, women will complete a pelvic floor muscle exercise diary. At 6 months, women will have a blinded assessment of their pelvic floor muscles to quantify pelvic floor muscle function.

Regarding the data collection tools, the International Consultation on Incontinence Questionnaire-Urinary Incontinence Short Form (ICIQ-UI SF), the ICIQ-Female Lower Urinary Tract Symptoms (FLUTS), the ICIQ-LUTSqol and the Pelvic Organ Prolapse Symptom Score (POP-SS) are instruments with evidence of reliability, validity and responsiveness. The Patient Global Impression of Improvement (PGI-I), the self-efficacy for PFMT scale and the EvQ-5D-3L have evidence of reliability and validity. The ICIQ Bowel Short Form used is an instrument not fully validated, but used because of its brevity and the lack of an alternative validated short bowel symptom questionnaire. The Oxford scale and International Continence Society (ICS) method of assessing pelvic floor function are standardised methods of digitally assessing the pelvic floor recognised by the International Continence Society. The remaining data are gathered via individual item questions focused on gathering specific information, for example, relating to whether a participant has received any other treatment for UI, and what incontinence products they have used.

### Primary outcome measure

The severity of UI at 24 months is the primary outcome, as measured by the validated ICIQ-UI SF which encompasses urinary leakage frequency, amount and interference with everyday life (total score 0–21, higher score indicates more severe UI).[21]

### Secondary outcomes measures

► Perceived improvement in UI (PGI-I).[22]
► Number with UI cured/improved (derived from the ICIQ-UI SF).[21]
► Uptake of surgery for UI.
► Uptake of other treatment for UI.
► Other lower urinary tract symptoms (eg, urgency, nocturia, pain, frequency) (ICIQ-FLUTS).[23]
► UI-specific quality of life (ICIQ-LUTSqol).[24]
► Prolapse symptoms (POP-SS).[25]
► Bowel symptoms (early version of ICIQ Bowel Short Form).
► Pelvic floor muscle function (Oxford scale,[26] ICS method[27]).
► Self-efficacy for PFMT (PFME self-efficacy scale).[28]
► Adherence to prescribed PFMT programme (derived from pelvic floor muscle exercise diary/follow-up questionnaire).
► Cost and use of NHS services.
► Cost to the women and their families/carers.
► Incremental cost per quality-adjusted life-year (QALY; EQ-5D-3L).[29]

### Centre selection

This trial will recruit from community, primary and secondary care settings across the UK. The first three centres will be identified and will start recruiting, providing an internal pilot. Experience of recruitment and data generated from these centres over 3 months will be used to fine-tune trial processes and protocols at new centres. Recruitment at other centres will be staggered with potentially more than 20 centres recruiting to the trial. Potential centres complete a feasibility questionnaire asking about: the number of women they could

recruit and how they would be identified; available staff, facilities and time to commit to the trial; previous experience of undertaking trials; any challenges and risks anticipated. A selection will be made on this basis.

## Participant selection

Women attending for a first continence appointment, or for a first outpatient appointment where UI is the presenting symptom, will be identified in advance from clinic lists in each centre, over the trial recruitment period. An OPAL Participant Information Leaflet will be given to potentially eligible women either before or at their routine appointment. When a woman attends for her appointment with her continence specialist they will confirm a clinical diagnosis of stress UI or mixed UI (those with urgency UI alone will be excluded), and assess other trial eligibility criteria by undertaking a vaginal examination as part of their routine care. If a woman is eligible and willing to take part, informed consent will be obtained by a member of the local research team.

### Inclusion criteria
► Women 18 years of age or above, presenting with a new episode of stress or mixed UI, who are willing and suitable to be randomised.

### Exclusion criteria
► Women who have urgency UI alone.
► Women who have had formal instruction in PFMT in the last year.
► Women who are unable to contract their pelvic floor muscles.
► Women who are pregnant or are less than 6 months postnatal.
► Women who have prolapse greater than stage II (>1 cm below the hymen on Valsalva).
► Women who are currently having treatment for pelvic cancer.
► Women who have cognitive impairment affecting capacity to give informed consent.
► Women who have neurological disease (multiple sclerosis, Parkinson's disease, stroke, motor neuron disease, spinal injury).
► Women with a known nickel allergy or sensitivity.
► Women who are currently participating in other research relating to their UI.

## Randomisation

Women who are eligible and willing to take part will be randomised to either the experimental or control group via the automated computer randomisation application developed at the CHaRT, University of Aberdeen. The web-based application will be accessed by staff at the centre to determine the participant's group. Allocation will be minimised on: type of UI (stress UI or mixed UI), centre, age (<50/≥50 years) and UI severity at baseline (ICIQ-UI SF score <13/≥13).[30]

## Masking

Statistical analysis will be conducted by research staff with the group allocation masked, using study identity numbers only to identify women and questionnaires. Due to the nature of the interventions, group allocation cannot be concealed from the women, the therapists delivering the intervention or those entering data. The clinician undertaking 6-month pelvic floor muscle assessments will not be aware of the woman's group allocation.

## Data collection and management

Data collection and entry is managed by experienced staff at the OPAL Trial Office situated in Nursing, Midwifery and Allied Health Professions (NMAHP) Research Unit, Glasgow Caledonian University. Data are entered into the CHaRT web-based database. The final trial dataset will be accessed only by the trial statisticians and health economists. Trained therapists or research nurses at centres collect data on paper Case Report Forms at every participant appointment. Centre staff enter the screening and eligibility data directly onto the web-based database in order to randomise participants. Participants complete paper diaries throughout treatment and outcome questionnaires at baseline, 6, 12 and 24 months' post-randomisation. Paper copies of case report forms, diaries and questionnaires are returned by post to the OPAL Trial Office to be entered onto the database. Copies of case report forms are also securely stored at each centre in accordance with local ethics protocols. Data from the longitudinal qualitative case study and process evaluation are gathered alongside and in addition to the main trial data and are described in the companion paper.

## Statistical analysis and sample size

All analyses will be according to the Statistical Analysis Plan and Health Economic Analysis Plan that will be agreed in advance with the Project Management Group (PMG) and the Trial Steering Committee (TSC). A single main analysis will be performed at the end of the trial when 24-month follow-up has been completed. An independent data monitoring and ethics committee (DMEC) will review confidential interim analyses of accumulating data at its discretion, at least annually. Analysis will be based on the intention-to-treat principle, such that women will be analysed in the groups to which they were allocated regardless of the intervention received.

The main effectiveness analysis will estimate the mean difference in the primary outcome (ICIQ-UI SF score) (and 95% CIs) between the experimental and control groups at 24 months using a general linear mixed model, adjusting for minimisation covariates, therapist type and baseline score. Centre and therapist type will be fitted, where possible, as random effects. Missing outcome data will be estimated using multiple imputation and included in a sensitivity analysis Secondary outcomes will be analysed in a similar manner using an appropriate generalised linear model. Complier average causal

effect analysis of the primary outcome will be conducted to investigate the impact of non-compliance. Subgroup analyses by type of incontinence (stress UI or mixed UI), type of therapist (physiotherapist or nurse), participant age (<50/≥50 years) and baseline UI severity (ICIQ-UI SF score <13/≥13) are also planned.

The primary economic outcome measure of cost effectiveness is incremental cost per QALY at 24 months undertaken from the NHS perspective. QALYs will be based on responses to the EQ-5D-3L[30] and the ICIQ-LUTSqol.[24] Personal patient resource use data will be incorporated into the analysis to report on a wider perspective. Where appropriate the analysis of incremental costs, effectiveness and cost-effectiveness will be based on similar statistical models as those outlined in the statistical analysis section above. This 'within' trial analysis will include both deterministic and stochastic sensitivity analyses to explore statistical and other forms (eg, around unit costs or the source of utility estimates) of uncertainty. Similar subgroup analysis will be performed in the economic analysis as defined in the statistical analysis if deemed relevant. If relevant an economic model will be developed to provide additional information for policy-makers.

Assuming a clinically meaningful difference of 3 points on the ICIQ-UI SF score (eg, change from leaking urine 'once a day' to 'never'), which is similar to the minimal clinically important difference of 2.5 reported in a study of older women,[31] and a conservative estimate for the SD of 10, a sample size of 234 per group would detect this difference (standardised effect size of 0.3) with 90% power at the 5% level of significance (two-sided alpha). Allowing for 22% drop out, we will randomise 300 women per group. With 600 subjects, we will also have 90% power to detect important differences in secondary outcomes.

## Safety measures

The OPAL trial involves treatments for UI which are well established in clinical practice, therefore adverse effects (although these are unlikely) will be those observed in everyday practice associated with the use of PFMT and biofeedback. In the OPAL trial, all serious adverse events experienced by a research participant will be reported to the main research ethics committee if they occur within 30 days of the participant's last PFMT appointment and where in the opinion of the Chief Investigator and/or the Chair of the DMEC the event is related and unexpected.

## Patient involvement

Women with experience of UI are involved in the design, delivery and oversight of the trial through membership on the PMG and TSC. A lay summary of the findings will be sent to participants involved in the trial.

## ETHICS AND DISSEMINATION

The trial will be conducted in accordance with the International Conference on Harmonisation Good Clinical Practice (ICH GCP)) Note for Guidance on Good Clinical Practice. Trial safety and progress will be overseen by the independent DMEC and the TSC. Meetings of these committees will take place annually.

Final trial results will be disseminated to the funding body, the NIHR Health Technology Assessment Programme. The trial results will then be submitted to peer-reviewed international academic journals and presented at international conferences. Results will also contribute to the relevant Cochrane review. Participants will be provided with a summary of the results.

**Author affiliations**
[1]Nursing, Midwifery and Allied Health Professions Research Unit, Glasgow Caledonian University, Glasgow, UK
[2]Faculty of Health Sciences and Sport, University of Stirling, Stirling, UK
[3]Department of Medicine, University of Otago, Dunedin, New Zealand
[4]Medical School, University of Exeter, Exeter, UK
[5]Health Services Research Unit, University of Aberdeen, Aberdeen, UK
[6]Division of Applied Health Sciences, University of Aberdeen, Aberdeen, UK
[7]Department of Obstetrics and Gynaecology, NHS Ayrshire and Arran University Hospital Crosshouse, Kilmarnock, UK
[8]School of Health and Life Sciences, Glasgow Caledonian University, Glasgow, UK
[9]Department of Gynaecology, NHS Greater Glasgow and Clyde, Glasgow, UK
[10]Edinburgh Clinical Trials Unit, University of Edinburgh, Edinburgh, UK
[11]Health Economics Research Unit, University of Aberdeen, Aberdeen, UK
[12]The Centre for Healthcare Randomised Trials, Health Services Research Unit, University of Aberdeen, Aberdeen, UK
[13]School of Nursing and Midwifery, Robert Gordon University, Aberdeen, UK
[14]Consumer Representative, Ayrshire, UK

**Acknowledgements** The authors also acknowledge the support of the National Institute for Health Research, through the Comprehensive Clinical Research Network. SD's position is partly supported by the National Institute for Health Research (NIHR) Collaboration for Leadership in Applied Health Research and Care South West Peninsula at the Royal Devon and Exeter NHS Foundation Trust.

**Contributors** SH directed the protocol development, inputting specifically to the trial design and outcome measurement components. JH-S, DM, SGD and JB developed the trial interventions, focusing on the behaviour change (JH-S, SD) and clinical (DM, JB) components. CG and JN contributed their expertise in clinical trial design, specific to the area of incontinence and complex intervention respectively. MA-f, WA and KG contributed their research and clinical expertise relating to urinary incontinence to inform the protocol development and its implementation. CB led the development of the process evaluation and qualitative longitudinal case study supported by SD and JH-S, ensuring that the longitudinal qualitative study and process evaluation protocol were congruent with and complementary to the trial protocol. AM and GM contributed to the development of the trial management and trial database processes respectively. MK developed the health economic methods within the protocol. AE developed the statistical methods within the protocol. LW contributed a consumer's perspective to the trial protocol as a whole. SS and NS contributed to the design of the implementation processes for trial management and data collection and management, respectively. AG contributed expertise in process evaluation to further develop those elements of the protocol.

**Funding** This work is supported by the NIHR Health Technology Assessment Programme (project reference: 11/71/03).

**Competing interests** None declared.

**Patient consent for publication** Not required.

**Ethics approval** Favourable ethics opinion covering recruitment across all UK NHS centres was obtained from the West of Scotland REC 4 (approved 13th March 2013, reference number 13/WS/0048) and local R&D departments.

**Provenance and peer review** Not commissioned; externally peer reviewed.

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
