## [Reviewer comments · BMJ Open]

This paper was submitted to a another journal from BMJ but declined for publication following peer review. The authors addressed the reviewers' comments and submitted the revised paper to BMJ Open. The paper was subsequently accepted for publication at BMJ Open.

(This paper received three reviews from its previous journal but only two reviewers agreed to published their review.)

ARTICLE DETAILS

TITLE (PROVISIONAL)	Effectiveness and cost-effectiveness of basic versus biofeedback-mediated intensive pelvic floor muscle training for female stress or mixed urinary incontinence: protocol for the OPAL randomised trial
AUTHORS	Hagen, Suzanne; McClurg, Doreen; Bugge, Carol; Hay-smith, Jean; Dean, Sarah Gerard; Elders, A; Glazener, Cathryn; Abdelfattah, Mohamed; Agur, Wael I; Booth, Jo; Guerrero, Karen; Norrie, John; Kilonzo, M; McPherson, Gladys; McDonald, Alison; Stratton, Susan; Sergenson, Nicole; Grant, Aileen; Wilson, Lyndsay

VERSION 1 – REVIEW

REVIEWER	Nathalia de Souza Abreu Freire Faculdade de Ciências Médicas e da Saúde de Juiz de Fora/MG (FCMS/JF), e Universidade Salgado de Oliveira, campus Juiz de Fora (UNIVERSO/JF).Brazil.
REVIEW RETURNED	05-Jun-2018

GENERAL COMMENTS	I suggest to mention in the introduction of the manuscript the good results that lombopelvic stabilization exercises have presented in the treatment of women with stress urinary incontinence when compared to exercises for the pelvic floor muscles, as evidenced by several studies (three referenced below and one attached). 1. Hung H-C, Hsiao S-M, Chih S-Y, Lin H-H, Tsauo J-Y. An alternative intervention for urinary incontinence: retraining diaphragmatic, deep abdominal and pelvic floor muscle coordinated function. Manual Therapy2010;15(3):273-9.2. Bø K, Mørkved S, Frawley H, Sherburn M. Evidence for benefit of transversus abdominis training alone or in combination with pelvic floor muscle training to treat female urinary incontinence: a systematic review. Neurourology and Urodynamics2009;28(5):368-73.3. Sapsford R, Hodges P, Smith M. Systematic review: Abdominal or pelvic floor muscle training. Neurourology and Urodynamics2010;29(5):800-1.
---

REVIEWER	Fátima F Fitz Federal University of São Paulo - Brazil
REVIEW RETURNED	17-Jun-2018

GENERAL COMMENTS	Comments to the Authors This study is of great relevance to the scientific community and it may also benefit the population that suffers from urinary incontinence. In the introduction, the authors discuss and present clearly the main issues related to PFMT and the addition of BF to PFMT for the treatment of SUI, as well as the strength and limitations of the study. However, I believe that authors can be able to present in more detail (method) what is being done. Following, I describe some considerations. In the ABSTRACT, the authors need to describe clearly the methods and analysis (type of the study, sample size, inclusion criteria, primary and secondary outcome, intervention period and follow-up). About the methodological description, aiming an improvement to a better methodological understanding, I suggest a presentation with the following topics: study design; study setting, recruitment and ethics; patient population; randomisation and blinding; interventions and comparison; outcome measures (primary and secondary outcomes); incidence of adverse events; data collection, management and monitoring; sample size estimates; and statistical analysis. In the protocol of the PFMT, the authors report that the number of repetitions, endurance maintenance and rest time, number of fast contractions were established according to the parameters of the PFM evaluation by the PERFECT system. The authors need to describe clearly in the example (page 7, line 21) which parameters of this system was used to elaborate the training program. It also needs to be reported if the patients receive some orientation regarding the position (supine, sit, orthostatic) to perform the exercise. Regarding the training of the counterbracing technique (also called 'The Knack'), the authors do not report how this training will be approached. So, I had some doubts: Were patients advised to perform the contraction before and during the daily stress activities? Was there a protocol for the patient perform the counterbracing technique? Was the home exercises the same that clinic protocol? Which BF equipment was used by the PFMT BF group: EMG or manometric? Auditory/visual feedback? In what position was performed the exercises? Did the BF group perform the counterbracing technique with BF device? Page 8, line 23-26: What is "CBTs"? How the "BCTs" was built? What are the components of the "BCTs"? it is necessary to describe in more details. Although the authors describe in Figure "THE OPAL TRIAL", I believe it is necessary to clarify in the text the intervention period, follow-up and in which moments the evaluations were be carried out. In the secondary outcome measures, the authors need to describe in more detail which each of them aimed to evaluate (improvement parameters). Ex. What was be considered adherence to the PFMT? What type of exercise diary was used? How the patients were instructed to fed the diary? How did the "follow-up questionnaire" assess the adherence? What are "Patient involvement"? Are patients who have urinary symptoms? Describe in detail the role of these women in the study. Why did the authors include this patient in the staff of the study?
---

REVIEWER	Priya Kannan The Hong Kong Polytechnic University Hong Kong
REVIEW RETURNED	19-Jun-2018

GENERAL COMMENTS	This paper has the potential to be accepted, however, some minor but important points have to be clarified before a positive action is taken  1. It is unclear from the paper if the outcome measures chosen are psychometrically sound. I suggest the authors add information about the validity and reliability of the chosen outcome measures. 2. Some sentences are too long and difficult to digest. For instance, under methods sections, lines 33-37 needs reconstruction for clarity. 3. I suggest the authors include information on the expertise of the therapist delivering the intervention. 4. Additional information on the biofeedback machine is required. Brand name, company, make etc. 5. Rationale for not screening participants based on bladder function test (cystogram, residual urine, urodynamic testing etc.) has to be reported?
--

VERSION 1 – AUTHOR RESPONSE

Reviewer's Comment	Authors' Response
Reviewer 1	
I suggest to mention in the introduction of the manuscript the good results that lombopelvic stabilization exercises have presented in the treatment of women with stress urinary incontinence when compared to exercises for the pelvic floor muscles, as evidenced by several studies (three referenced below and one attached).  1. Hung H-C, Hsiao S-M, Chih S-Y, Lin H-H, Tsauo J-Y. An alternative intervention for urinary incontinence: retraining diaphragmatic, deep abdominal and pelvic floor muscle coordinated function. Manual Therapy2010;15(3):273-9. 2. Bø K, Mørkved S, Frawley H, Sherburn M. Evidence for benefit of transversus abdominis training alone or in combination with pelvic floor muscle training to treat female urinary incontinence: a systematic review. Neurourology and Urodynamics2009;28(5):368-73. 3. Sapsford R, Hodges P, Smith M. Systematic review: Abdominal or pelvic floor muscle training. Neurourology and Urodynamics2010;29(5):800-1. 	The focus of the OPAL trial is the effect of adding biofeedback to PFMT, rather than the effect of PFMT in its own right. Therefore we feel it would be confusing to add information about the evidence comparing PFMT with another exercise approach i.e. lombopelvic stabilization exercises.
Reviewer 2	
Comments to the Authors This study is of great relevance to the scientific community and it may also benefit the population that suffers from urinary incontinence. In the introduction, the authors discuss and present clearly the main issues related to PFMT and the addition of BF to PFMT for the	

treatment of SUI, as well as the strength and limitations of the study. However, I believe that authors can be able to present in more detail (method) what is being done. Following, I describe some considerations.

In the ABSTRACT, the authors need to describe clearly the methods and analysis (type of the study, sample size, inclusion criteria, primary and secondary outcome, intervention period and follow-up).

About the methodological description, aiming an improvement to a better methodological understanding, I suggest a presentation with the following topics: study design; study setting, recruitment and ethics; patient population; randomisation and blinding; interventions and comparison; outcome measures (primary and secondary outcomes); incidence of adverse events; data collection, management and monitoring; sample size estimates; and statistical analysis.

We have kept to the 300-word limit and headings in the Abstract required by the journal, therefore it is not possible to include more detail here. All the information mentioned by the reviewer is however in the main paper. Some of the information suggested is already in the abstract, and we could add some additional information (in yellow) relating to the reviewer's suggestions if the editors are happy to allow a slightly longer Abstract (366 words):

- Type of the study/study design – we state “multicentre randomised controlled trial”
- Sample size – we state “Six hundred women will be randomised”
- Inclusion criteria/patient population – although we do not list all inclusion criteria, we say “women with stress UI or mixed UI”
- Primary and secondary outcome – although we do not list secondary outcomes, we state “The primary outcome is UI severity at 24 months after randomisation”
- Intervention period and follow-up/interventions and comparison – we suggest adding “All participants are offered six PFMT appointments over 16 weeks. The use of clinic and home biofeedback is added to PFMT for participants in the biofeedback group.”
- Study setting – we could make the following change “Six hundred women *from UK community, outpatient and primary care settings* will be randomised.”
- Recruitment and ethics – details of the ethics approval are included in the Abstract.
- Randomisation and blinding – could add “Group allocation could not be masked from participants and healthcare staff.”

	 • Incidence of adverse events – this manuscript described the trial protocol. The trial is not complete therefore the incidence of AEs would not be reported here. We state “Serious adverse events will be reported to the data monitoring and ethics committee, the ethics committee and trial centres as required.” • Data collection, management and monitoring – we do not feel that these details would be appropriate in the abstract. We suggest adding “ ..will be randomised and followed up via questionnaires, diaries and pelvic floor assessment.” • Statistical analysis – we state “An intention-to-treat analysis of the primary outcome will estimate the mean difference between the trial groups at 24 months using a general linear mixed model adjusting for minimisation covariates and other important prognostic covariates, including the baseline score.”
In the protocol of the PFMT, the authors report that the number of repetitions, endurance maintenance and rest time, number of fast contractions were established according to the parameters of the PFM evaluation by the PERFECT system. The authors need to describe clearly in the example (page 7, line 21) which parameters of this system was used to elaborate the training program. It also needs to be reported if the patients receive some orientation regarding the position (supine, sit, orthostatic) to perform the exercise.	The therapist uses the information from PERFECT regarding Power (P), Endurance (E) and Repetitions (R) to determine a starting dose comprised of the number of maximum contractions a woman could do in a row (based on R) and held for how long (based on E), to take the muscle to the point of ‘fatigue’ (“every contraction timed” (ECT) component of PERFECT). The number of repetitions per set (R) will usually be between 1 and 10, and the length of hold (E) between 1 and 10 seconds. This will be progressed over time to reach 3 sets of 10 contractions held for 10 seconds each (and 10 sec rest) to increase power (P). Regarding fast contractions (F), the therapists are instructed that fast contractions can be included, if indicated, as part of a PFMT programme. The primary indications are training a fast response to a change in intra-abdominal pressure (‘The Knack’) or training a swift response to the sensation of urgency. Therefore, this is at the discretion of the therapist in terms of how many they add, or if they do this at all, and when.

	The therapists are instructed that lying, sitting and standing positions can be used at their discretion based on the findings of the pelvic floor muscle assessment in conjunction with participant preference. Exercise position may be selected to minimise co-contraction in unwanted muscle groups and/or used to progress the exercise programme (e.g. from gravity assisted to gravity-resisted exercise positions). The BCT checklists ask the therapist to record body positions used as a behaviour change technique, making the behavioural skill more difficult over time by including more gravity resisted (e.g. upright) positions. We have added a sentence to clarify this: “The starting dose comprises the number (and duration of hold) of near maximal contractions a woman is able to complete before the pelvic floor muscles reached fatigue.”
Regarding the training of the counterbracing technique (also called 'The Knack'), the authors do not report how this training will be approached. So, I had some doubts: Were patients advised to perform the contraction before and during the daily stress activities? Was there a protocol for the patient perform the counterbracing technique? Was the home exercises the same that clinic protocol?	Each therapist is given an Intervention Manual which covers all aspects of delivering the trial interventions. The material in the Intervention Manual is also covered in the training day attended by all therapists. The therapists are instructed to teach ‘The Knack’ using a voluntary cough as the ‘test’ activity. The pelvic floor muscles are contracted before the cough, held during the cough, and released after the abdominal/diaphragmatic contraction associated with coughing ceases. The functionality of ‘The Knack’ is progressed with the introduction of other appropriate activities (e.g. lifting, pushing, pulling, bending down/over, getting up from ground level) as clinically indicated. We have added a sentence to clarify this: “Counterbracing is taught with a cough – a pre-contraction of the pelvic floor muscles to be held until the abdominal/diaphragmatic contraction ceases – and over time progressed to include other functional activities as clinically indicated.” The exercise programme including “The Knack” is practised during the appointment. This determines the home exercise

	programme goal which is agreed with the woman and written in her pelvic floor exercise diary. The use of “The Knack” is encouraged in between appointments.
Which BF equipment was used by the PFMT BF group: EMG or manometric? Auditory/visual feedback? In what position was performed the exercises? Did the BF group perform the counterbracing technique with BF device? Page 8, line 23-26: What is “CBTs”? How the “BCTs” was built? What are the components of the “BCTs”? it is necessary to describe in more details.	EMG biofeedback is used in the PFMT BF group. The participant has visual and auditory feedback from biofeedback device. Verbal feedback from the therapist to the woman based on digital vaginal palpation is permitted in both trial groups. Manometric feedback is not used. Similarly, electrical stimulation is not permitted. The PFMT protocol allows lying, sitting and standing exercise positions. The position(s) recommended to the participant are at the therapist’s discretion based on the pelvic floor muscle assessment in conjunction with participant preference. The therapists teach women to use counterbracing with any activity which causes increased intra-abdominal pressure. Selection of activities is at the discretion of the therapist based on the woman’s clinical history and their assessment. Counterbracing is not recommended whilst using the BF device. This is a typographical error and should say BCTs. This has been corrected here and in two other places. The BCTs to be used are listed in the therapist’s notes as a checklist so that she can indicate which BCTs are used during each appointment. There are 30 core BCTs and 17 optional BCTs so it is not possible to include details of them all in the paper but this could be provided as supplementary material if allowable. The main categories of BCTs included in the interventions are: beliefs, emotions and information; teaching and confirming PFM contraction, practising skills; goal setting; action planning; problem solving. We have added this summary information to the manuscript.
Although the authors describe in Figure "THE OPAL TRIAL", I believe it is necessary to clarify in the text the intervention period, follow-up and in which moments the evaluations were be carried out.	The intervention period is 16 weeks with appointments at weeks 0, 1, 3, 6, 10 and 15 (page 6, line 39).

In the secondary outcome measures, the authors need to describe in more detail which each of them aimed to evaluate (improvement parameters). Ex. What was be considered adherence to the PFMT? What type of exercise diary was used? How the patients were instructed to fed the diary? How did the “follow-up questionnaire” assess the adherence?	A section on Data Collection has been added to clarify when outcomes were measured. Some detail has been added to explain further what is being evaluated by the secondary outcome measures. The exercise diary was developed for the study in a booklet format. It includes a first page where details of the exercise programme are written by the therapist, a second page of completion instructions for the woman and an example, and 4 further pages with blank rows, one for each day. A new diary is provided by the therapist at each appointment, and the woman returns her previously completed diary to the therapist at the next appointment. For each day the woman enters the date, number of exercise sessions undertaken (with and without use of BF if she is in the BF group) and any comments she wishes to make. The follow up questionnaires include 2 questions addressing adherence:  - Have you done any pelvic floor muscle exercises over the last month? (yes/no) - If yes, how often did you do the exercises? (a few times a month/a few times a week/once a week/once a day/a few times a day)
What are "Patient involvement"? Are patients who have urinary symptoms? Describe in detail the role of these women in the study. Why did the authors include this patient in the staff of the study?	Women who have experienced urinary incontinence themselves and its treatment were asked to be part of the trial in various ways. One woman agreed to be part of the study team who applied for the grant, to strengthen the quality and patient-centredness of the project by ensuring its aims and methods would be relevant and acceptable to women with UI. This approach to Patient and Public Involvement is encouraged by the funding body who fund the study (NIHR HTA). Another woman agreed to be a member of the Trial Steering Committee which oversees the trial as it progresses. Thus she is able to contribute to the committee’s role of advising

	on the continuation of the trial and how to address any challenges met, and ensuring compliance with the planned protocol, from the perspective of a patient.
Reviewer 3	
This paper has the potential to be accepted, however, some minor but important points have to be clarified before a positive action is taken 1. It is unclear from the paper if the outcome measures chosen are psychometrically sound. I suggest the authors add information about the validity and reliability of the chosen outcome measures.	The ICIQ-UI SF, the ICIQ-FLUTS, the ICIQ-LUTSqol, the POP-SS are instruments with evidence of reliability, validity and responsiveness. The PGI-I, the Self-Efficacy for PFMT scale and the EQ-5D-3L have evidence of reliability and validity. The ICIQ Bowel Short Form used is an instrument not fully validated, but used because of its brevity and the lack of an alternative validated short bowel symptom questionnaire. The Oxford scale and ICS method of assessing pelvic floor function are standardised methods of digitally assessing the pelvic floor recognised by the International Continence Society. The remaining outcomes are individual item questions gathering specific information, e.g. relating to whether a participant had received any other treatment for UI, and what incontinence products they had used. This information is given in the new Data Collection section.
2. Some sentences are too long and difficult to digest. For instance, under methods sections, lines 33-37 needs reconstruction for clarity.	We have re-written this sentence and 2 others as follows:  • This trial design is a parallel group multicentre randomised controlled trial, comparing effectiveness and cost-effectiveness of PFMT versus biofeedback-mediated PFMT for women with stress or mixed UI. It is set in UK community, outpatient and primary care settings, where continence care is usually provided. The trial includes a nested longitudinal qualitative case study and process evaluation (Figure 1). • The programme is progressed (e.g. increasing the number of repetitions by one, or duration of hold by one second, each week) according to the woman's ability to reach her individualised goal. For example, a goal might be to exercise 3 days per week. On these days the women could aim for 3 exercise sets, with 3 minutes between sets. Each set could comprise 10 repetitions held for 10 seconds each, with 10 seconds rest

	between, followed by 10 fast contractions.  Biofeedback also supports motivation through providing audio and visual feedback (graphs on screen which can be printed off) on the 'effort/performance' of the muscle contraction, which facilitates tracking changes in muscle strength and performance. In turn this enhances behavioural skills through improving the performance of a muscle contraction, and the timing of a contraction, to reduce leakage with increases in intra-abdominal pressure (e.g. during cough, sneeze, lift).
3. I suggest the authors include information on the expertise of the therapist delivering the intervention.	Therapists delivering the intervention are clinical specialists or advanced practitioners already working in the area of continence. This information has been added to the manuscript.
4. Additional information on the biofeedback machine is required. Brand name, company, make etc.	All biofeedback machines used in the study, both in clinic and at home, are the same type: Neurotrac Simplex, Verity Medical Limited. However, we are reluctant to publish this information as we do not want to support a particular product. We would prefer to describe this simply as a hand-held EMG biofeedback device with Bluetooth technology which measures and monitors pelvic floor muscle contractions.
5. Rationale for not screening participants based on bladder function test (cystogram, residual urine, urodynamic testing etc.) has to be reported?	This is a pragmatic trial, recruiting women who had already had a diagnosis of urinary incontinence, and who in clinical practice would have been offered PFMT. Some of these women may have had previous investigations but they were not required to establish eligibility for this trial.

VERSION 2 – REVIEW

REVIEWER	Fátima F. Fitz Federal University of São Paulo
REVIEW RETURNED	30-Sep-2018
GENERAL COMMENTS	I thank the authors for reviewing the article and for the replies sent. I agree with the changes and justifications presented and I have no further considerations to make.